# Estimation of Fatty Acids in Intramuscular Fat of Beef by FT-MIR Spectroscopy

**DOI:** 10.3390/foods10010155

**Published:** 2021-01-13

**Authors:** María José Beriain, Francisco C. Ibañez, Edurne Beruete, Inmaculada Gómez, Miguel Beruete

**Affiliations:** 1Instituto de Innovación y Sostenibilidad en la Cadena Agroalimentaria, Universidad Pública de Navarra, 31006 Pamplona, Spain; pi@unavarra.es (F.C.I.); eduberujor@hotmail.com (E.B.); 2Departamento de Biotecnología y Ciencia de los Alimentos, Universidad de Burgos, 09001 Burgos, Spain; igbastida@ubu.es; 3Instituto de Investigación Sanitaria de Navarra (IdiSNA), Universidad Pública de Navarra (UPNA), Irunlarrea 3, 31008 Pamplona, Spain; miguel.beruete@unavarra.es

**Keywords:** FT-MIR spectroscopy, beef, *n*-3 fatty acids, prediction models

## Abstract

The aim of this research was to estimate the fatty acid (FA) content of intramuscular fat from beef by Fourier transform mid-infrared (FT-MIR) spectroscopy. Four diets were supplemented in 10% linseed (LS) and/or 2% conjugated linoleic acid (CLA): CON (without L or CLA), LS, CLA, and LS+CLA. For each diet, 12 young Holstein bulls were allocated. The spectral response of the beef samples was analyzed applying FT-MIR spectroscopy (from 400 to 4000 cm^−1^) and predictive models were developed using partial least square regression with cross-validation. The obtained coefficients (*R*^2^) for some FA, such as α-linolenic acid with a *R*^2^ = 0.96 or *n*-3 polyunsaturated fatty acids (*n*-3 PUFA) with *R*^2^ = 0.93, demonstrate that FT-MIR spectroscopy is a valid technique to estimate the content of FA. In addition, samples were correctly classified according to the animal diet using discriminant analysis in the region 3000–1000 cm^−1^. The obtained results suggest that the FT-MIR spectroscopy could be a viable technique for routine use in quality control because it provides fast and sustainable analysis of FA content. Furthermore, this technique allows the rapid estimation of the FA composition, specifically *n*-3 PUFA and CLA, of nutritional interest in meat. It also allows the classification of meat samples by the animal diet.

## 1. Introduction

In recent years, the demand for high quality and safety in food production and development for new products enriched with bioactive compounds with health promoting properties has grown. Therefore, the food industries require appropriate analytical tools to satisfy this demand. Some of the features demanded for these technologies are that they are fast to perform, easy to apply, and that they require a simple manipulation of the samples, alongside the avoidance of samples destruction, waste minimization, and low cost [1].

Fat is a critical component of meat because it has a great influence on the maintenance of muscular tissue reducing protein breakdown and it is the energy storage reservoir. In addition, intramuscular fat is responsible for the organoleptic properties and a necessary component of meat products. Fat contributes to and influences palatability, tenderness, juiciness, and flavor of meat. Currently, the tendency of the meat industry is the modification of the lipid profile of meat products, by reducing the saturated fatty acids (SFA) content and increasing the *n*-3 and *n*-6 polyunsaturated fatty acids (PUFA) which are considered essential to maintain the health [2]. Moreover, several studies have revealed that conjugated linoleic acid (CLA) and some of the *n*-3 PUFA provide beneficial effects to human health. For instance, α-linolenic acid (ALA), eicosapentaenoic acid (EPA), and docosahexaenoic acid (DHA) present potential anti-inflammatory properties against diseases, such as obesity or diabetes [3]. Regarding CLA, studies in animal models suggest protective effects against obesity and atherosclerosis [4].

Meat composition is influenced by numerous factors. Animal diet is particularly interesting as it is a factor easy to manipulate and still has an important effect on its composition [5]. Thus, several dietary intervention trials have been conducted to enhance the *n*-3 PUFA and the CLA content in beef [6]. Therefore, increasing these fatty acids (FA) in the meat is one of the greatest open scientific challenges. One of the alternatives can be by means of dietary supplementation of PUFA. One of the most widely used natural sources in *n*-3 PUFA is flax or linseed. The linseed coat provides protection to FA against biohydrogenation by ruminal microorganisms and thus facilitates the duodenal passage of PUFA [7]. Thus, linseed supplementation to bulls has been considered by several authors [8,9] who observed increments of ALA, EPA, and docosapentaenoic acid (DPA) proportions. Likewise, Gillis et al. [10,11] and Schegel et al. [12] investigated other dietary interventions, based on the addition of the CLA in rumen protected form in order to avoid biohydrogenation of CLA.

In order to assess the effectiveness of the different dietary *n*-3 PUFA supplementations, the FA profile changes must be evaluated in meat. That requires an appropriate method for extraction of fat from meat. This extraction has to be performed with minimal exposure to heat and light to prevent the modification of *n*-3 PUFA and CLA and to prevent changes in FA structure, which reduces the nutritional value of fats. Several techniques have been employed to analyze FA profile, gas chromatography (GC) being the most commonly used one. However, this method is not satisfactory, as it requires a lot of sample preparation and a lot of processing time. To overcome the limitations of classical chemical methods, alternative techniques based on physical methods have been sought. Mid-infrared (MIR) spectroscopy arises as an interesting alternative due to its high speed of analysis and environmental sustainability as no harmful substances for the environment are used [13].

The determination of the FA profile in different types of meat products has been carried out by different authors using Fourier-transform mid-infrared (FT-MIR) spectroscopy. For example, Ripoche and Guillard [14] and Flatten et al. [15] studied the profile of FA of some samples of pork fat and the proportions of DPA and DHA in pork fat, respectively. Moreover, the fat content is a parameter that is usually controlled during meat production. Thus, fast and trustworthy techniques would help the meat industry to determine lipid content. In this sense, a recent study [16] for determining the lipid content of meat samples from different species showed a good accuracy (R^2^ = 0.9173) using FT-MIR spectroscopy. Likewise, Ruiz et al. [17] reported that this technique could be a suitable technique to differentiate meat from “old vs. young” foals.

Currently, the challenge in the meat industry is to assess PUFA contents because they are quality attributes of meat products associated with consumers’ health. Fast and environmentally sustainable methods are required for predicting minority PUFA content in a reliable way, especially *n*-3 PUFA of nutritional interest such as DHA, EPA, DPA, and CLA. The conventional method of extraction is based on the use of chemical solvents (chloroform-methanol) by multiple steps that requires several hours for the FA determination under study. Instead, FT-MIR spectroscopy could be an alternative technique to determine this type of FA of nutritional interest in meat, with or without prior removal of intramuscular fat. Therefore, the aims of this work are (i) to check the appropriate sample (meat or extracted fat) to predict the FA content in beef by FT-MIR spectroscopy; (ii) to discriminate beef samples with different contents in *n*-3 PUFA and CLA using FT-MIR spectroscopy.

## 2. Materials and Methods

### 2.1. Animal Management and Meat Sampling

Beef samples were obtained from 48 Holstein bulls fed with one of four dietary treatments (12 animals per dietary treatment). Animals diets were isoenergetic and isoproteic and differed in their amount of whole linseed (LS) and rumen-protected CLA (Lutrell^©^ pure, BASF, Ludwigshafen, Germany): CON (without LS or CLA), 10% LS, 2% CLA, and LS+CLA. Composition of the experimental diets, animal productive performance and carcass characteristics of the animals used in this study have been reported by Albertí et al. [18]. Animals were cared for in accordance with EU Directive [19]. Finishing period was reached at 123.0 ± 11.2 days, and then the animals (live weight 458.4 ± 16.6 kg) were slaughtered at an EU-licensed commercial abattoir following standard procedures. At 24 h after slaughter, *Longissimus thoracis* steaks (100 g) were cut at the sixth rib level from the left half-carcass, vacuum-packaged and frozen, and stored at −20 °C until analyses. Samples were gently thawed at 4 °C overnight prior to analyze.

### 2.2. Extraction of the Intramuscular Fat of the Meat Samples

The Soxhlet method was used to extract the intramuscular fat of the meat samples [20].

### 2.3. Mid-Infrared Spectra Measurements and Spectral Acquisition

Meat samples were analyzed by direct FT-MIR spectroscopy. In total, five replicates were performed per meat sample analyzed. Besides, the intramuscular fat extracted from all samples was also analyzed by FT-MIR spectroscopy (Figure 1). Two extractions of fat per animal were made. Each fat sample was analyzed in duplicate.

A Fourier-transform infrared (FTIR) Vertex 80v spectrometer (Bruker Optik GmbH, Ettlingen, Germany) was used to obtain the infrared spectra. The configuration used in the experiments was a Globar IR thermal source (operation bandwidth, 6000–50 cm^−1^), a KBr beamsplitter (10,000–400 cm^−1^), and a DLaTGS detector (10,000–250 cm^−1^). An A225/Q Platinum Attenuated Total Reflectance (ATR) accessory was used to get all measurements. A calibration was done before each experiment by measuring the response without any sample on the ATR. Each sample was placed on the ATR touching the diamond crystal and 32 scans in the 4000–400 cm^−1^ spectral range were recorded with a resolution of 4 cm^−1^. After the measurements, a data analysis was performed to select the wavenumbers of the peaks with higher intensities of absorption and to calculate then the standard deviation between the pair of absorption intensities.

### 2.4. Data Analysis

A specific program of chemometrics, OPUS Quant 2 (Bruker Optik GmbH, Ettlingen, Germany) was used for building the models. The reference method to develop the regression equations and estimate the content of FA in this work was carried out by GC. Fatty acid profile was determined in previous research [21]. For the current study, the most relevant FAs from the nutritional point of view were selected. Table A1 shows the intramuscular FA content in muscle from young Holstein bulls fed with different diets.

MIR spectral data were preprocessed by applying the first derivative and vector normalization. Then calibration models between FA values and MIR spectra were computed by partial least square (PLS) regression and validated using cross-validation. Prediction residuals were then combined to calculate the root mean square error of cross-validation (RMSECV) [22,23]. It is known that the main advantage of cross validation is that it requires a low number of samples because the method is calibrated and validated by the same group of samples. In this method, one sample is excluded from the group of samples before starting the calibration and the rest is used to calibrate the method. In this way, the cycle is repeated retiring a different sample each time until all samples have taken part for validation once. A principal component analysis (PCA) was performed before PLS regression models were developed to determine any relevant and interpretable structure in the data and to detect sample outliers. The optimal number of terms in the PLS calibration models was specified by the lowest number of factors associated with the minimum value of RMSECV in order to avoid overfitting the models [22]. Statistics calculated for the calibrations included the coefficient of determination in cross validation (*R*^2^), ratio of performance to deviation (RPD), and bias.

A spectral region between 2800 and 2300 cm^−1^ was omitted from PLS analysis, due to uncertainty in that range, which may be the consequence of the absorption of CO_2_.

The statistical software SPSS 23.0 (IBM Corp., Armonk, NY, USA) was used to analyze the fat spectral information, performing a stepwise discriminant analysis (test of goodness of fit of independent variables by Lambda Wilks; *p* ≤ 0.05). The aim of these analyses was to determine the feasibility of classifying beef samples of the same diet together.

## 3. Results and Discussion

As Gomez et al. [21] found, the diets affect the FA content of beef samples. The content of ALA in bulls fed following LS and LS+CLA diets were 6-fold higher than those fed with CON and CLA diets (12.89 vs. 1.96 mg/100 g muscle; *p* < 0.001). The diet enriched with *n*-3 PUFA led to increases of *n*-3 PUFA. The amount of ALA varies depending on the diet. This fatty acid is EPA and DPA precursor, and there is a relation between ALA and these fatty acids. When ALA amount in intramuscular fat is high, there are more EPA and DPA.

### 3.1. FTIR Spectra Measurement and Assignment of Representative Bands

Figure A1 shows a representative spectrum of the fat samples from Holstein bulls with the characteristic peaks and the assignment with chemical functional groups obtained by FT-MIR analysis from 4000 to 400 cm^−1^. The assignment of the most important bands was done by matching the wavenumbers with the bibliography references. Table 1 contains the principal band assignment.

A broad band with low intensity appears around 3005 cm^−1^. This band is lined up with the C-H bond vibration of the unsaturated fatty acids, stretching vibration of *cis* double bond (C=CH) [24,25,26,27]. The signals at 2920 and 2854 cm^−1^ is related to asymmetric and symmetric stretching vibration of C-H bonds present on methylene (CH_2_) and methyl (CH_3_) groups in fatty acids [14,24,28].

Between 1800 and 400 cm^−1^, there is a narrow peak around 1743 cm^−1^ that stands out from the rest with a higher absorption intensity than the others. This peak is related to the stretching vibration of carbonyl bond of esters and free FA [14,24,25,26,27,29]. Lower absorption intensities were shown by the rest of the bands that appear in the spectra outlined above. Nevertheless, they provide information. The band around 1465 cm^−1^ is related to scissoring bending vibration mode of C-H bond in the methyl and methylene groups [24,25,26,28,29,30,31]. The band at 1160 cm^−1^ is associated to stretching vibration of C-O bonds and bending vibration of C-H bonds [24,28,31]. Finally, around 721 cm^−1^ there is a band which is related to the overlapping of the methylene (CH_2_) rocking vibration and the out of plane vibration of *cis*-disubstituted olefins [14,24,25,26,27].

Figure 2 shows the mean absorbance spectra of 240 beef samples according to the four dietary treatments. Figure 3 shows the mean absorbance spectra of 96 fat samples according to the four dietary treatments. As shown there, all spectra have the same peaks at the same wavenumbers, revealing that all the samples have the same bonds. Therefore, the differences among the different diets can only be found in the absorption intensity of each peak.

The spectra of Figure 2 and Figure 3 are different and by comparing them it is noted that the spectra from intramuscular fat extracted from beef gives more information. A possible explanation for this is that the water and the rest of components of beef could mask the spectra of FA.

Figure 4 shows the spectral characteristics of the fat samples in three specific ranges (3000–2800, 1800–1700, 1500–1000 cm^−1^). At first sight, the CON sample spectra differ noticeably with respect to the other samples. This indicates that the method is able to detect beef enriched by *n*-3 PUFA and/or CLA.

### 3.2. Prediction Models

Partial least square (PLS) was used for constructing the prediction models. The model validation was done using cross validation, taking out one sample each time.

The PLS analysis was done using the data of GC as reference values. The determination coefficients (*R*^2^) obtained for each prediction model and the errors are shown in Table 2. The *R*^2^ value informs how close the CG values and FT-MIR predictive values are. The values varied between 0.96 and 0.15 for the different FA. The main conclusion is that MIR spectroscopy can be used to predict some FA sums (i.e., ∑*n*-3 PUFA, ∑CLA, ∑MUFA, ∑PUFA, and ∑SFA) but also for individual PUFA (i.e., ALA, DPA, EPA, RA, C18:1*t*10+C18:1*t*11). The determination coefficients found in this study are lower than the results obtained in pork [13,14]. Ripoche and Guillard [14] reported higher determination coefficients for ∑SFA (0.92), ∑MUFA (0.98), and ∑ PUFA (0.98) in pork fat by FT-MIR spectroscopy. The lower validation coefficients of the present study (∑SFA = 71.12; ∑MUFA = 76.85; ∑PUFA = 71.95), can be explained because ruminants transform FA in the rumen, and the amount and composition of fat leaving the rumen differ from intake [32].

ALA was the best predicted FA (*R*^2^ = 96.21%). This result could be explained because the difference in the content among the different groups was higher than the other FA analyzed by GC. Gómez et al. [20] reported that samples from bulls fed with L and L+CLA diets were around 8–9-fold higher than those from bulls fed with CON and CLA diets (Table A1). This FA is critical because it is the precursory of DHA and EPA and those PUFA are important because they have an effect on the maintenance of normal brain function and normal vision, and on the maintenance of normal cardiac function [33]. In addition, the group that had the best *R*^2^ (94.65%) was the sum of *n*-3 PUFA. The worst *R*^2^ was obtained on the prediction of the CLA*t*10,*c*12 FA, which had a value of 15.48%. Although the results obtained in this work are interesting to estimate the FA in intramuscular fat, more studies are needed to improve the prediction in the content of other fatty acids such as C18:2*n*-6(LA), C18:0, and C18:1*c*9, and especially DHA.

Figure 5 shows the scatter plot for the relationship between the reference value and predicted value for the main FA that are linked to positive health effects (ALA, EPA, DPA, and DHA).

ALA is the precursor of EPA, and this in turn of DPA and DHA sequentially. However, conversion from ALA is very inefficient in humans (below 5–10%). This is one reason why the diet should provide a sufficient amount of all these *n*-3 PUFA [34]. On the other hand, *n*-3 PUFA are highly susceptible to oxidation during processing and storage of foods. This oxidation negatively affects the quality of meat by producing off-flavors and reducing their nutritional value. It is even suspected that free radicals derived from PUFA may be responsible for certain diseases [35]. Therefore, MIR spectroscopy could allow a quick analysis of the meat composition and ensure its quality.

### 3.3. Discriminant Analysis

The last objective of this study was to classify the meat samples from animals fed with different dietary treatments (CON, LS, CLA, and LS+CLA), so stepwise discriminant analysis was employed. The spectral data from 3000 to 1000 cm^−1^ obtained in the fat extracted from beef samples were used.

The classification matrix of the discriminant analysis is shown in Table 3. When all samples were taken together, 92.5 % of the cases were correctly classified. This indicates a good agreement between the real grouping of the animals and the assignment into the groups by discriminant analysis method.

Figure 6 shows the plot of the two canonical functions for the selected spectral range. Canonical function 1 indicated that the centroid of CON group was located on the left side of the plot. The centroid of LS and CLA fell near the center, whereas the centroid LS+CLA group was located on the right side. Moreover, function 2 discriminated the groups because the centroids of C and LS+CLA groups were located on the top side, whereas the CLA and LS were located on the bottom side. Finally, LS+CLA group showed a more homogenous distribution than the rest of the groups.

Therefore, four groups of animal samples were separated depending on the diet provided (CON, LS, CLA, and LS+CLA) in terms of their spectra obtained by FT-MIR. These results evidenced that discriminant analysis can be used to explore the relationship between FA composition and the spectra of beef from bulls and their diets. Moreover, earlier works using discriminant analysis gave good results to characterize beef FA profile [36], to distinguish veal of weaned and unweaned calves [37], to separate meat samples according to aging time [38], to differentiate between four different diets to feed cattle during the finishing period [39], or to classify beef samples depending on the finishing diet fed to bulls [40].

## 4. Conclusions

The present study confirmed the potential of FT-MIR technique for the rapid and nondestructive measurement of several intramuscular FA from beef. In general, the work related to the determination and quantification of the FA composition of the meat uses the conventional method of extraction employing an apolar solvent as it is less aggressive for GC. This requires several stages of extraction with chloroform-methanol using chemical solvents and several hours for the determination of the FAs under study. Even though the method precedes a fat extraction (6 h) the infrared method is faster because it requires only 1 min per sample in comparison to GC method (1 h). Additionally, infrared spectroscopy is easier to undertake and low cost. Furthermore, this technique allows the rapid estimation of the FA composition, specifically *n*-3 PUFA and CLA, of nutritional interest in meat. Finally, discriminant analysis allows the classification of samples by the animal diet. In the absence of previous studies on intramuscular beef fat, the MIR technique allows an estimation of the content of FA in the intramuscular fat of beef. For that, the fat is extracted by a standardized method. This analytical technique can be applied for the quality control of beef, especially in case of nutritional interest *n*-3 PUFA and CLA, being a rapid and sustainable method.

## Figures and Tables

**Figure 1 foods-10-00155-f001:**
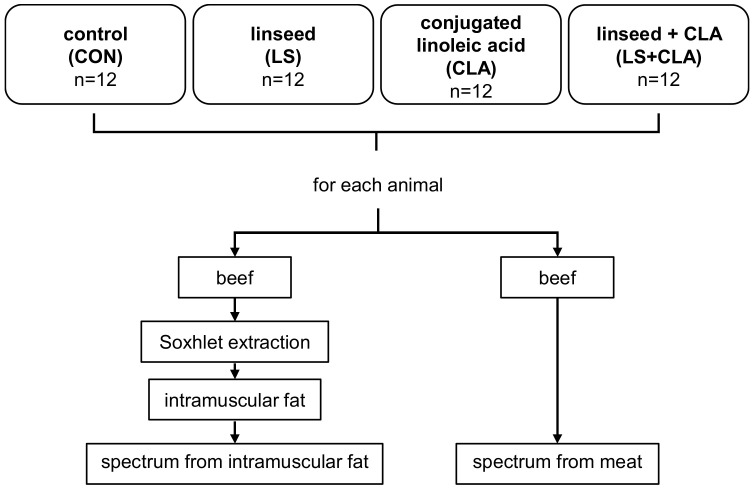
Experimental design.

**Figure 2 foods-10-00155-f002:**
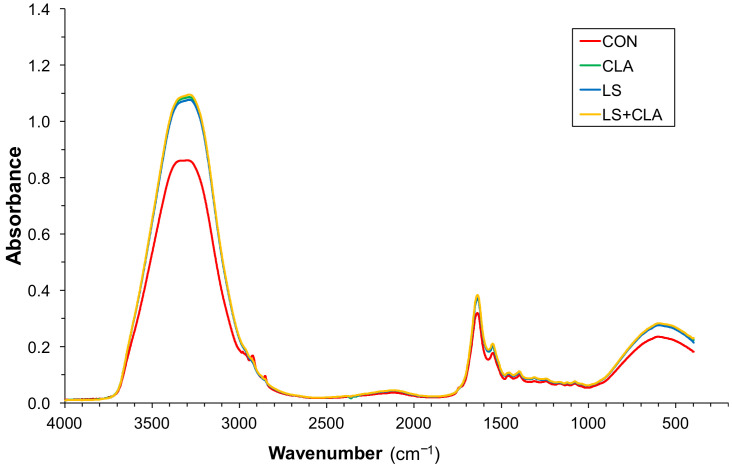
Mean absorbance spectra of 240 beef samples according to the four dietary treatments: CON = control; LS = 10% linseed; CLA = 2% conjugated linoleic acid; LS+CLA.

**Figure 3 foods-10-00155-f003:**
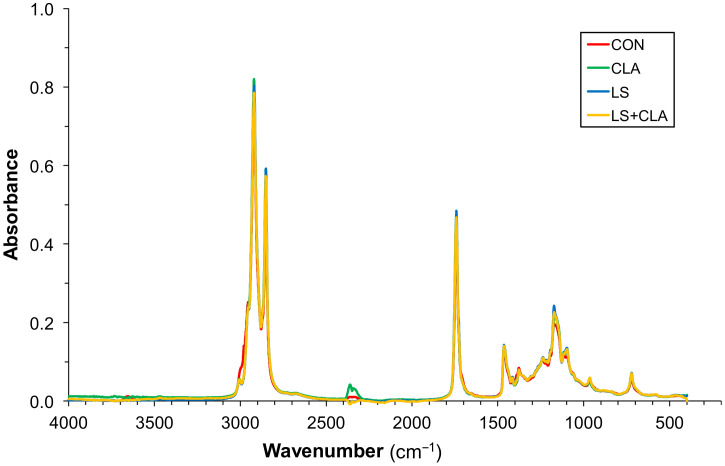
Mean absorbance spectra of 96 fat samples according to the four dietary treatments: CON = control; LS = 10% linseed; CLA = 2% conjugated linoleic acid; LS+CLA.

**Figure 4 foods-10-00155-f004:**
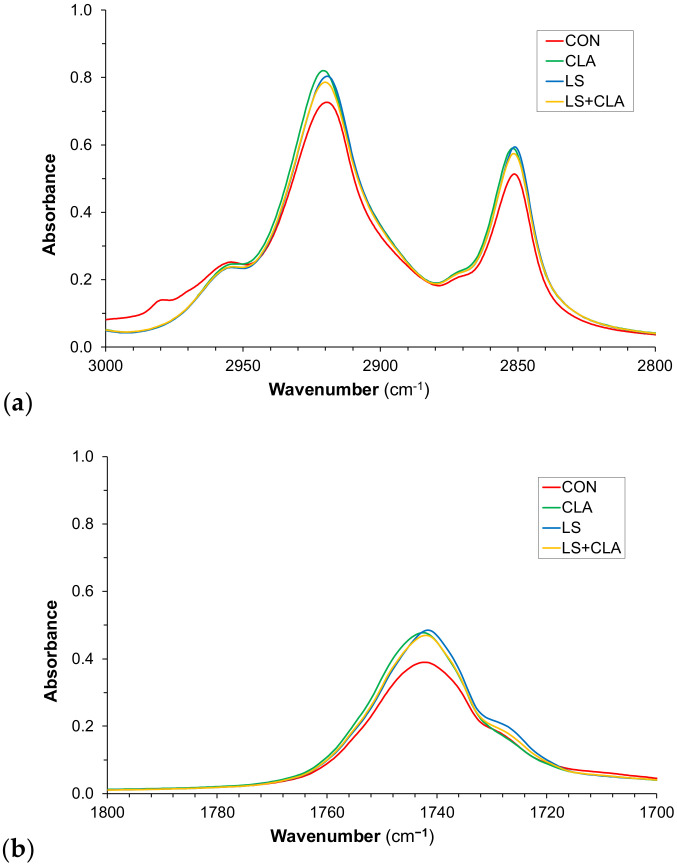
Average fat FTIR-spectra for fat samples according to the four dietary treatments: CON = control, LS = 10% linseed, CLA = 2% conjugated linoleic acid, and LS+CLA. Range selected from 3000 to 2800 cm^−1^ (**a**), from 1800 to 1700 cm^−1^ (**b**), and from 1500 to 1000 cm^−1^ (**c**).

**Figure 5 foods-10-00155-f005:**
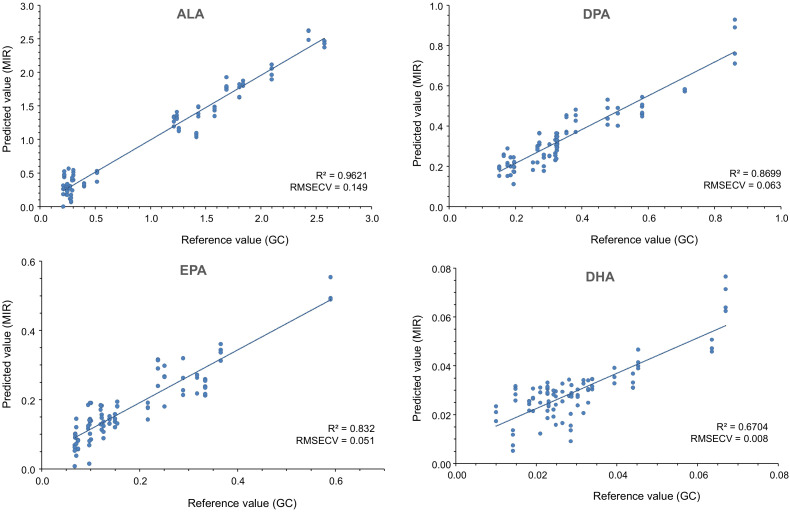
Scatter plot showing the relationship between the reference value, as determined by gas chromatography, and the predicted value in α-linolenic acid (ALA), docosapentaenoic acid (DPA), eicosapentaenoic acid (EPA), and docosahexaenoic acid (DHA) contents by MIR spectroscopy; *R*^2^ = coefficient of determination. RMSECV = root mean square error of cross-validation.

**Figure 6 foods-10-00155-f006:**
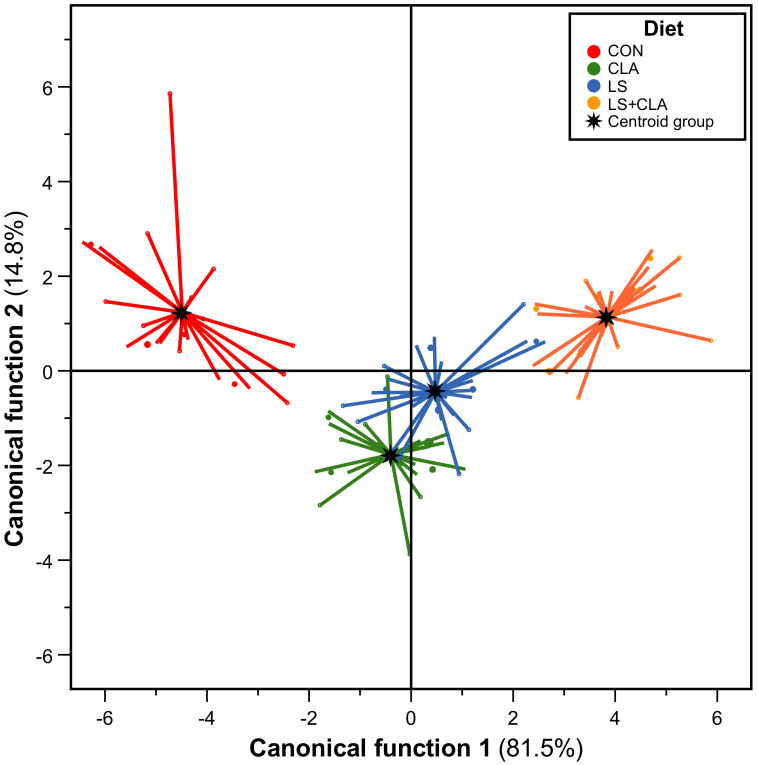
Graphic representation of the fat samples from young Holstein bulls fed with the four different diets (CON, LS, CLA, and LS+CLA) according to stepwise discriminate analysis including spectra of fat samples. Lines connect each sample with group centroid. Range selected from 3000 to 1000 cm^−1^.

**Table 1 foods-10-00155-t001:** Wavenumber assignment with the chemical functional groups.

Wavenumber (cm^−1^)	Functional Group	References
3005	CH stretching vibration in cis double bond =CH; unsaturated fatty acids	[24,25,26,27]
2920	C-H asymmetric stretching of CH_2_ and CH_3_; aliphatic groups	[14,24,25,26,27]
2854	C-H symmetric stretching of CH_2_ and CH_3_; aliphatic groups	[14,24,25,26,27,28]
1743	C=O stretching of esters: free fatty acids	[14,24,25,26,27,29]
1465	C-H scissoring vibration	[24,25,26,28,30,31]
1377	Symmetric bending vibrations of CH_3_ groups	[24,25,26,27]
1160	C-O stretching vibration y C-H bending	[24,28,31]
721	Overlapping of the methylene (-CH_2_) rocking vibration and to the out of plane vibration of *cis*-disubstituted olefins	[14,24,25,26,27]

**Table 2 foods-10-00155-t002:** Optimized model of the fatty acid content of beef: coefficients of determination (*R*^2^) and root mean square error for cross-validation (RMSECV) and for calibration (RMSEC).

Fatty Acid	Intramuscular Fat Extracted from Beef				Beef			
	Validation		Calibration		Validation		Calibration	
	*R* ^2^	RMSECV	*R* ^2^	RMSEC	*R* ^2^	RMSECV	*R* ^2^	RMSEC
C16:0	78.73	0.94	88.55	0.72	1.30	40.20	7.02	39.30
C18:0	62.47	0.89	83.26	0.63	33.38	23.60	41.27	22.70
C18:1c9	53.85	2.16	82.52	1.41	32.99	28.40	53.58	24.60
C18:1*t*10+C18:1*t*11	81.17	0.35	91.29	0.25	28.35	8.12	47.60	7.13
C18:2*n*-6 (LA)	59.63	1.80	82.58	1.23	34.45	15.60	46.73	14.60
C18:3*n*-3 (ALA)	96.21	0.15	98.68	0.09	48.72	4.67	68.94	3.78
CLA*c*9,*t*11 (RA)	82.03	0.04	87.80	0.03	48.09	1.58	66.81	1.30
CLA*t*10,*c*12	15.48	0.01	78.07	0.006	28.33	0.17	55.09	0.14
C20:4*n*-6	57.49	0.64	77.04	0.49	25.36	8.08	51.90	6.72
C20:5*n*-3 (EPA)	82.44	0.05	93.82	0.03	37.07	1.61	57.34	1.38
C22:5*n*-3 (DPA)	86.61	0.06	95.34	0.04	35.41	3.41	52.82	3.04
C22:6*n*-3 (DHA)	66.64	0.007	82.48	0.006	17.1	0.15	33.63	0.14
Ʃ*n*-6	57.72	2.53	82.98	1.70	29.02	23.80	47.98	21.00
Ʃ*n*-3	94.65	0.24	98.01	0.16	45.75	8.73	61.79	7.64
ƩCLA	80.31	0.06	90.10	0.04	36.67	2.97	53.93	2.60
ƩSFA	71.12	1.17	85.35	0.89	18.66	68.30	26.26	66.30
ƩMUFA	76.85	1.83	93.63	1.03	32.42	29.20	53.64	25.20
ƩPUFA	71.95	2.33	88.13	1.60	39.44	29.10	49.95	27.40

*R*^2^ = coefficient of determination; RMSECV = root mean square error of cross-validation; RMSEE = root mean square error of estimation Σ*n*-6: sum of C18:2*n*-6*t*9,*t*12, C18:2*n*-6, CLA*t*10,*c*12,C18:3*n*-6, C20:3*n*-6, C20:4*n*-6, and C22:4*n*-6. Σ*n*-3: sum of C18:3*n*-3, C20:5*n*-3, C22:5*n*-3, and C22:6*n*-3. ΣCLA: sum of CLA*c*9,*t*11,CLA*t*10,*c*12,CLA*c*9,*c*11, and CLA*t*9,*t*11. ∑SFA = total saturated fatty acids; ∑MUFA = total monounsaturated fatty acid; ∑PUFA = total polyunsaturated fatty acids.

**Table 3 foods-10-00155-t003:** Classification matrix in percentage using the spectral information of the fat samples scanned from 3000 to 1000 cm^−1^. Correctly classified cases are in diagonal marked in bold.

		Classified into Group			
		CON	CLA	LS	LS+CLA
**Actual group**	CON	**95.2**	4.8	0.0	0.0
	CLA	0.0	**95.8**	4.2	0.0
	LS	0.0	16.7	**79.2**	8.3
	LS+CLA	0.0	0.0	0.0	**100.0**

Control (CON); 10% linseed (LS); 2% CLA (CLA); 10% L + 2% CLA (LS+CLA).

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
