# Peer review of "Estimation of Fatty Acids in Intramuscular Fat of Beef by FT-MIR Spectroscopy"

_foods, 2021, doi:10.3390/foods10010155_

Round 1
Reviewer 1 Report
Dear authors,
I read very carefully the article " Mid-Infrared Spectroscopy for Quantification of Fatty Acids in Beef Enriched by n-3 PUFA and Conjugated Linoleic Acid" which I found interesting and in line with the purposes of the magazine.
The manuscript is well written and it points out the importance to differentiate beef, with different contents in n-3 PUFA and CLA, and to test the adequacy of the FT-MIR spectroscopy technique to predict the contento of these fatty acids in the intramuscular fat of beef. The materials and methods section and the part of the statistics are well addressed. My only main concern is the use of data (table 1) that you have already published in a previous work (Gomez et al., ref 19). From this consideration arise some questions to you for evaluating together the opportunity of publishing such data.
In details:
line 45: the reference 2, from my point of view, is inappropriate; please select a more fitting reference.
Figure 1; please review the caption;
section 2.2; if the GC data are adapted from Gómez et al. [19] , this section should be not reported in the experimental section.
section 2.3. please describe how the intramuscolar fat was extracted for FT-MIR experiment and if there are some difference from intramuscolar fat extracted for GC experiments. If there are any, please explain why two different extractive methods were used forintramuscolar fat. If there are no differences, please add in Figure 1 that the extracted fat has been used for both determinations (GC anf MIR-FTIR).
Table 1: how the data was adapted from your previou work (Gómez et al. [19])? In your opinion is mandatory to report these data or it is only enougth to mention them by reference [19] ?
line 253. "human species"
In the conclusion section , the pros and cons of such application should be reported. (e.g. for which fatty acids the methodology does not apply).
Author Response
I read very carefully the article " Mid-Infrared Spectroscopy for Quantification of Fatty Acids in Beef Enriched by n-3 PUFA and Conjugated Linoleic Acid" which I found interesting and in line with the purposes of the magazine.
The manuscript is well written and it points out the importance to differentiate beef, with different contents in n-3 PUFA and CLA, and to test the adequacy of the FT-MIR spectroscopy technique to predict the content of these fatty acids in the intramuscular fat of beef. The materials and methods section and the part of the statistics are well addressed. My only main concern is the use of data (table 1) that you have already published in a previous work (Gomez et al., ref 19). From this consideration arise some questions to you for evaluating together the opportunity of publishing such data.
Thank you very much for your helpful comments. The authors have revised the paper accordingly and want to highlight that your comments contribute to improve our manuscript. Please find our response (in blue) to reviewer’s specific comments (in black) below.
Authors have improved the background and have included the following relevant references in the introduction:
[2] De Smet, S.; Vossen, E. Meat: The balance between nutrition and health. A review. Meat Sci 2016, 120, 145–156, doi:10.1016/j.meatsci.2016.04.008.
[3] SokoÅ‚a-WysoczaÅ„ska, E.; WysoczaÅ„ski, T.; Wagner, J.; Czyż, K.; Bodkowski, R.; LochyÅ„ski, S.; Patkowska-SokoÅ‚a, B. Polyunsaturated Fatty Acids and Their Potential Therapeutic Role in Cardiovascular System Disorders—A Review. Nutrients 2018, 10, 1561, doi:10.3390/nu10101561.
[4] den Hartigh, L.J. Conjugated Linoleic Acid Effects on Cancer, Obesity, and Atherosclerosis: A Review of Pre-Clinical and Human Trials with Current Perspectives. Nutrients 2019, 11, doi:10.3390/nu11020370Dilzer, A.; Park, Y. Implication of Conjugated Linoleic Acid (CLA) in Human Health. Crit Rev Food Sci Nutr 2012, 52, 488–513, doi:10.1080/10408398.2010.501409
The methods have been adequately described as it was shown in section 2. Materials and Methods and the changes have made according with your suggestions. The conclusions have also modified to be supported by the results. In detail:
- Line 45: the reference 2, from my point of view, is inappropriate; please select a more fitting reference.
The reference 2 has been replaced.
- Figure 1; please review the caption;
Figure 1 has been revised and simplified as “Experimental design”.
- Section 2.2; if the GC data are adapted from Gómez et al. [19] , this section should be not reported in the experimental section.
The mention to GC method from experimental section 2.2. (Gómez et al. [19]) has been deleted and Table 1 has been moved to Appendix section. Also, we have introduced a new line in the data analysis.
- Section 2.3. please describe how the intramuscular fat was extracted for FT-MIR experiment and if there are some difference from intramuscular fat extracted for GC experiments. If there are any, please explain why two different extractive methods were used for intramuscular fat. If there are no differences, please add in Figure 1 that the extracted fat has been used for both determinations (GC and MIR-FTIR).
For the FT-MIR method, fat was extracted with the Soxhlet technique according to Whittington et al. (1986). This technique facilitates the complete removal of fat. For the GC method, fat was extracted according to Kramer et al. (1998). This technique is appropriate for previous derivatization of free fatty acids according to Aldai et al. (2010).
- Aldai N, Dugan ME, Kramer JK, Robertson WM, Juárez M, Aalhus JL. Trans-18:1 and conjugated linoleic acid profiles after the inclusion of buffer, sodium sesquicarbonate, in the concentrate of finishing steers. Meat Sci. 2010;84(4):735-41. doi: 10.1016/j.meatsci.2009.11.009.
- Kramer JK, Sehat N, Dugan ME, Mossoba MM, Yurawecz MP, Roach JA, Eulitz K, Aalhus JL, Schaefer AL, Ku Y. Distributions of conjugated linoleic acid (CLA) isomers in tissue lipid classes of pigs fed a commercial CLA mixture determined by gas chromatography and silver ion-high-performance liquid chromatography. Lipids. 1998 33(6):549-58. doi: 10.1007/s11745-998-0239-1.
- Whittington, F.M., Prescott, N.J., Wood, J.D. and Enser, M. The effect of dietary linoleic acid on the firmness of backfat in pigs of 85 kg live weight. Sci. Food Agric., 1986, 37: 753-761. https://doi.org/10.1002/jsfa.2740370807.
- Table 1: how the data was adapted from your previous work (Gómez et al. [19])? In your opinion is mandatory to report these data or it is only enough to mention them by reference [19] ?
As the authors have previously commented they have eliminated the GC data adapted from Gómez et al. [now 20] from experimental section 2.2. and Table 1 has been moved to the Appendix section.
- Line 253. "human species" ¿???
Word “species” has been deleted.
- In the conclusion section, the pros and cons of such application should be reported. (e.g. for which fatty acids the methodology does not apply).
We have added this paragraph in the conclusion section: “This study provides a new method for estimating the fatty acid content in the intramuscular fat in beef. This fat is extracted using a conventional technique (Soxhlet technique). This method can be adapted for the quality control of beef.”

Reviewer 2 Report
Manuscript foods-1038344 reports on the quantification of Fatty Acids in Beef Enriched by n-3 PUFA and Conjugated 3 Linoleic Acid using FT-MIR spectroscopy.
In my opinion this is clearly a confirmatory study with very limited, if any, novelty. The use of FT-MIR spectroscopy for the same purpose has been previously used (ref. 13, 14, 23, 27 ). The same holds for the addition of conjugated linoleic acid or linoleic acid for the improvement of meat nutritional value (ref. 9, 19) as well as for the use of fatty acid analysis to discriminate diets fed to cattle (ref. 38, 39). Based on these facts, the authors should clearly state what new is presented in the present study.
Another problem is that the title does not reflect all the work done. i.e. since one of the study objectives was to discriminate bull meat according to diet, this should be included in the manuscript title.
Thirdly, the text should be checked for the proper use of English by a native English speaker.
My detailed comments follow the text sequence:
l.25: add ‘it’ before ‘is’ in the sentence
l.26: change ‘MIR’ to ‘FT-MIR’
l.34: change ‘of’ to ‘for’
l.35: change ‘biocompounds’ to ‘bioactive compounds’
l.36: change ‘require’ to ‘requires’
l.39-40: how does fat contribute to the preservation of meat ?
l.40: delete word ‘the’ before word ‘responsible’
l.42: delete word ‘to’
l.43: add word ‘the’ before ‘meat industry’
l.45: change ‘essentials’ to ‘essential’
l.49-50: rewrite sentence in proper English
l.55: change ‘interesting’ to ‘of scientific interest’
l.58: delete ‘the’ after ‘thus’
l.63-65: rewrite sentence in proper English
l.67: text should read:… changes in FA structure which reduces (or downgrades) the nutritional value of fats.
l.68-70: rewrite sentence in proper English
l.71: delete word ‘the’
l.72: add ‘it’ after ‘because’
l.73: change ‘substance’ to ‘substances’
l.74: delete ‘The’ at the beginning of sentence
l.79: rewrite sentence in proper English
L.91: change ‘entire males’ to ‘bulls’
l.104: delete ‘Summarize the experimental design of present study’. Retain: Experimental design’
l.107: delete ‘as is’
l.110: change ‘Total fat method’ to ‘The Soxhlet method’
l.113: delete word ‘analysis’
l.114-115: rewrite sentence in proper English
l.166 and Table 1: how were these data adapted to be used in the present study ?
l.218: delete ‘the’ before ‘gas chromatography’
l.219: change ‘models’ to ‘model’
Τable 3: with only 2 exceptions (96.21 and 94.65) the R2 value is statistically speaking very low. How does this fact affect discrimination of diets ?
l.236: delete ‘The’
l.241: change ‘in’ to ‘on’
l.245: delete ‘The’
l.252: add ‘the’ before ‘precursor’
l.256 and 258: change ‘their’ to ‘its’
l.260: delete ‘by’
l.295: the authors speak of FT-MIR as though they have applied it for the first time ! See my general comments on the previous use of this technique by numerous researchers.
Based on the above, I recommend major revision expecting the authors to clarify the real novelty of the work.
Author Response
Dear Authors, I consider the study very interesting and well structured. However, I suggest you to review the language and to provide the following improvements
Thank you very much for your helpful comments. The authors have revised the paper accordingly and want to highlight that your comments have contributed to improve the manuscript. Please find our response (in blue) to reviewer’s specific comments (in black) below.
The English style and grammar in the present manuscript have been revised as recommended by the reviewer.
The results have been modified to improve their presentation according with your comments.
The conclusions have also changed to supported by the results
- In my opinion this is clearly a confirmatory study with very limited, if any, novelty. The use of FT-MIR spectroscopy for the same purpose has been previously used (ref. 13, 14, 23, 27). The same holds for the addition of conjugated linoleic acid or linoleic acid for the improvement of meat nutritional value (ref. 9, 19) as well as for the use of fatty acid analysis to discriminate diets fed to cattle (ref. 38, 39). Based on these facts, the authors should clearly state what new is presented in the present study.
Authors have made some changes in the introduction taking into account the reviewer’s comments. In this way, authors agree that FT-MIR spectroscopy as well the addition of conjugated linoleic acid or linoleic acid for the improvement of meat nutritional value and the use of fatty acid analysis to discriminate diets fed to cattle have been previously used for the same purpose. Nevertheless, it should be noted that there are no previous studies on intramuscular beef fat by FT‑MIR spectroscopy. There are studies on beef tallow. Other work carried out on beef focuses on detecting adulterations by the use of mixtures.
There are studies on pork in which the fat is extracted and analyzed by MIR and GC. Two methods have been followed: the subcutaneous fat is melted prior to analysis by MIR or is extracted with hexane for analysis by GC.
The novelty of this work is based on the method to estimate the AG content in the intramuscular fat of beef (extracted with the Soxhlet method). If a conventional method employing an apolar solvent was used, it is less aggressive for FA, but several stages of extraction with chloroform-methanol would be required. This requires the use of several chemical solvents and several extractions for the determination of the AGs.
- Another problem is that the title does not reflect all the work done. i.e. since one of the study objectives was to discriminate bull meat according to diet, this should be included in the manuscript title.
Authors agree with the reviewer so the title has changed as follows: “Estimation of Fatty Acids in Intramuscular Fat of Beef by FT-MIR Spectroscopy”
- Thirdly, the text should be checked for the proper use of English by a native English speaker.
The English style and grammar in the present manuscript have been revised as recommended by the reviewer.
- My detailed comments follow the text sequence:
l.25: add ‘it’ before ‘is’ in the sentence
The change has been made.
l.26: change ‘MIR’ to ‘FT-MIR’
The change has been made.
l.34: change ‘of’ to ‘for’
The change has been made.
l.35: change ‘biocompounds’ to ‘bioactive compounds’
The change has been made.
l.36: change ‘require’ to ‘requires’
The change has been made.
l.39-40: how does fat contribute to the preservation of meat ?
The sentence has been improved as follows: “Fat is one of the most important components of meat because it has a great influence on the maintenance of muscular tissue reducing protein breakdown preservation of muscle and it is the energy storage reservoir”.
l.40: delete word ‘the’ before word ‘responsible’
The change has been made
l.42: delete word ‘to’
The change has been made
l.43: add word ‘the’ before ‘meat industry’
The change has been made
l.45: change ‘essentials’ to ‘essential’
The change has been made
l.49-50: rewrite sentence in proper English
The sentence has been rewritten in proper English.
l.55: change ‘interesting’ to ‘of scientific interest’
The change has been made
l.58: delete ‘the’ after ‘thus’
The change has been made
l.63-65: rewrite sentence in proper English
The diet enriched with linseed led to increases of n-3 PUFA. The amount of ALA varies depending on the diet. This fatty acid is EPA and DPA precursor, and there is a relation between ALA and these fatty acids. When ALA amount in intramuscular fat is high, there are more EPA and DPA.
l.67: text should read:… changes in FA structure which reduces (or downgrades) the nutritional value of fats.
The change has been made
l.68-70: rewrite sentence in proper English
Sentence has been rewritten
l.71: delete word ‘the’
The change has been made
l.72: add ‘it’ after ‘because’
The change has been made
l.73: change ‘substance’ to ‘substances’
The change has been made
l.74: delete ‘The’ at the beginning of sentence
The change has been made
l.79: rewrite sentence in proper English
Sentence has been rewritten
L.91: change ‘entire males’ to ‘bulls’
The change has been made
l.104: delete ‘Summarize the experimental design of present study’. Retain: Experimental design’
The change has been made
l.107: delete ‘as is’
The change has been made
l.110: change ‘Total fat method’ to ‘The Soxhlet method’
The change has been made
l.113: delete word ‘analysis’
The change has been made
l.114-115: rewrite sentence in proper English
Sentence has been rewritten
l.166 and Table 1: how were these data adapted to be used in the present study ?
Fatty acid profile was determined in a previous research [reference 20]. For the current study, the most relevant FAs from the nutritional standpoint were selected. Table A1 show the intramuscular FA content in muscle from young Holstein bulls fed with different diets (page 15, Appendix).The FA contents were the variable to predict from spectral data obtained by FT-MIR. Authors have introduced an explication for clarifying how these data were adapted to be used in the present study.
l.218: delete ‘the’ before ‘gas chromatography’
The change has been made.
l.219: change ‘models’ to ‘model’
The change has been made.
Τable 3: with only 2 exceptions (96.21 and 94.65) the R2 value is statistically speaking very low. How does this fact affect discrimination of diets ?
Authors agree with the reviewer that R2 value for C18:3n-3 (ALA) and Ʃn-3 estimation are the highest, but there are other R2 values for C22:5n-3 (DPA) and C20:5n-3 (EPA) and almost the rest of fatty acids (FA) that could be considered as good enough to estimate the fatty acid composition. Besides, discriminant analysis was useful to separate successfully the beef FA profile from four types of diet fed cattle.
l.236: delete ‘The’
The change has been made
l.241: change ‘in’ to ‘on’
The change has been made
l.245: delete ‘The’
The change has been made
l.252: add ‘the’ before ‘precursor’
The change has been made
l.256 and 258: change ‘their’ to ‘its’
The change has been made
l.260: delete ‘by’
The change has been made
l.295: the authors speak of FT-MIR as though they have applied it for the first time! See my general comments on the previous use of this technique by numerous researchers.
Based on the above, I recommend major revision expecting the authors to clarify the real novelty of the work.
Thank you very much for your help. Authors think that have clarified the novelty of the work and have introduced an explanation in the lines.

Round 2
Reviewer 2 Report
In the revised text (1038344- R1), the authors have addressed most of my comments. My main question remains regarding the novelty of the study. In their response, the authors state:
‘authors agree that FT-MIR spectroscopy as well the addition of conjugated linoleic acid or linoleic acid for the improvement of meat nutritional value and the use of fatty acid analysis to discriminate diets fed to cattle have been previously used for the same purpose. Nevertheless, it should be noted that there are no previous studies on intramuscular beef fat by FT‑MIR spectroscopy. There are studies on beef tallow. Other work carried out on beef focuses on detecting adulterations by the use of mixtures’.
My question is: what is the important difference in analyzing beef tallow vs. intramuscular beef fat ? Later on they state:
‘The novelty of this work is based on the method to estimate the AG content in the intramuscular fat of beef (extracted with the Soxhlet method)’.
My question is : is the Soxhlet extraction of fat from meat muscle a novel procedure ? A clear answer regarding the novelty of the study should be provided in the text.
To another comment of mine regarding R2 the authors answer as follows:
‘Authors agree with the reviewer that R2 value for C18:3n-3 (ALA) and Æ©n-3 estimation are the highest, but there are other R2 values for C22:5n-3 (DPA) and C20:5n-3 (EPA) and almost the rest of fatty acids (FA) that could be considered as good enough to estimate the fatty acid composition’.
My question is: are R2 values 78.73 %(for C16:0), 62.47 %(C18:0), 53.85 %(for C18:1c9), 59.63 %[for C18:2n-6(LA)], 15.48 %(for CLAt10,c12), 57.49 %(for C20:4n-6), 66.64 %[for C22:6n-3 (DHA)] considered satisfactory ?
Finally, minor editing type errors in the text should be corrected. i.e.
l.33: change ‘healthy’ to ‘health benefiting’ or ‘health promoting’
l.76: change ‘So’ to ‘Thus’
Author Response
Comments and Suggestions for Authors
Thank you very much for your helpful comments. Please find our response (in blue) to reviewer’s specific comments (in black) below.
In the revised text (1038344- R1), the authors have addressed most of my comments. My main question remains regarding the novelty of the study. In their response, the authors state:
‘authors agree that FT-MIR spectroscopy as well the addition of conjugated linoleic acid or linoleic acid for the improvement of meat nutritional value and the use of fatty acid analysis to discriminate diets fed to cattle have been previously used for the same purpose. Nevertheless, it should be noted that there are no previous studies on intramuscular beef fat by FT‑MIR spectroscopy. There are studies on beef tallow. Other work carried out on beef focuses on detecting adulterations by the use of mixtures’.
My question is: what is the important difference in analyzing beef tallow vs. intramuscular beef fat ?
Thank you very much for your helpful suggestions about the novelty of this work. The authors' response to the reviewer is based on Kauffman et al., (2012) and Lopez-Bote (2017). Authors consider that the PUFA analyzed in this paper (DHA, EPA, DPA and n-3 PUFA totals) require a specific extraction technique. Phospholipids are located in the cell membranes that contain a high proportion (above 30%) of essential polyunsaturated fatty acids. On the other hand, the beef tallow is a fatty deposit located in the subcutaneous and intermuscular region. It is visible and easy to remove. Their FAs are mostly found as triacylglycerides. The n-3 PUFAs studied (DPA, DHA and EPA) are minority compared to the other AFs in the beef tallow composition.
- Kauffman, R.G. Meat Composition. In Handbook of Meat and Meat Processing (2nd edition); Hui, Y.H., Ed.; CRC Press: Boca Raton (USA), 2012; pp. 45–61 ISBN 978-0-429-15146-0.
- López-Bote, C. Chemical and Biochemical Constitution of Muscle. In Lawrie´s Meat Science (8th edition); Toldrá, F., Ed.; Woodhead Publishing Series in Food Science, Technology and Nutrition; Woodhead Publishing: Duxford (UK), 2017; pp. 99–158 ISBN 978-0-08-100694-8.
Later on they state:
‘The novelty of this work is based on the method to estimate the AG content in the intramuscular fat of beef (extracted with the Soxhlet method)’.
The authors apologize to reviewer 2 because they have not been able to adequately explain the novelty of this work and they greatly appreciate this opportunity. The novelty of this study is in the rapid determination of minority AF such as n-3 PUFA and CLA fatty acids, since they are essential nutrients for a diet to be considered health promoting. The extraction of these PUFA requires gentle handling conditions and techniques based on gas chromatography with the development of estimation equations by using pure standards. This requires a longer extraction time, separation and very time-consuming analytical identification and long analysis times. The technique based on the FT-MIR method to estimate the content of this nutritional interest PUFA is the novelty of this work.
My question i : is the Soxhlet extraction of fat from meat muscle a novel procedure ? A clear answer regarding the novelty of the study should be provided in the text.
A sentence regarding the novelty of the study has been provided in the text:
- Abstract: line 27-28
- Introduction: lines 85- 87
- Conclusions: 282-284
To another comment of mine regarding R2 the authors answer as follows:
‘Authors agree with the reviewer that R2 value for C18:3n-3 (ALA) and Æ©n-3 estimation are the highest, but there are other R2 values for C22:5n-3 (DPA) and C20:5n-3 (EPA) and almost the rest of fatty acids (FA) that could be considered as good enough to estimate the fatty acid composition’.
My question is: are R2 values 78.73 %(for C16:0), 62.47 %(C18:0), 53.85 %(for C18:1c9), 59.63 %[for C18:2n-6(LA)], 15.48 %(for CLAt10,c12), 57.49 %(for C20:4n-6), 66.64 %[for C22:6n-3 (DHA)] considered satisfactory ?
Authors recognized that R2 value for others fatty acids are not so good that the R2 value for DPA, EPA or n-3 PUFA but the result for the fatty acids of nutritional interest could justify adequately or enough the advance of knowledge of using the FT-MIR spectroscopy. Conventional techniques such as CG involve more difficulties to manipulate this fatty acids minority because they can be damaged by chemical changes in their configuration.
Finally, minor editing type errors in the text should be corrected. i.e.
Authors thank very much for your helpful corrections in the editing type
l.33: change ‘healthy’ to ‘health benefiting’ or ‘health promoting’
Authors have done it
l.76: change ‘So’ to ‘Thus’
Authors have done it
